# Limited Probiotic Effect of *Enterococcus gallinarum* L1, *Vagococcus fluvialis* L21 and *Lactobacillus plantarum* CLFP3 to Protect Rainbow Trout against Saprolegniosis

**DOI:** 10.3390/ani13050954

**Published:** 2023-03-06

**Authors:** Juan-Miguel Fregeneda-Grandes, Concepción González-Palacios, Tania Pérez-Sánchez, Daniel Padilla, Fernando Real, José-Miguel Aller-Gancedo

**Affiliations:** 1Departamento de Sanidad Animal, Campus de Vegazana s/n, Universidad de León, 24071 León, Spain; 2Departamento Agricultura y Alimentación, Universidad de La Rioja, Avenida Madre de Dios 53, 26006 Logroño, Spain; 3Instituto Universitario de Sanidad Animal y Seguridad Alimentaria (IUSA), Universidad de Las Palmas de Gran Canaria, Carretera de Trasmontaña s/n, 35416 Araucas, Spain

**Keywords:** biocontrol, probiotics, saprolegniosis, inhibition assays, *Oncorhynchus mykiss*

## Abstract

**Simple Summary:**

Probiotics have been considered as alternatives to the antibiotics currently used to control diseases caused by different microorganisms, which show high prevalence and losses in aquaculture. In this study, the possible utility of three probiotics (*Enterococcus gallinarum* L1, *Vagococcus fluvialis* L21 and *Lactobacillus plantarum* CLFP3—effective against vibriosis or lactococosis in sea bass or rainbow trout) was investigated for the biocontrol of saprolegniosis in rainbow trout. For this purpose, both in vitro inhibition studies and competition for binding sites against *Saprolegnia parasitica* and in vivo tests with experimentally infected rainbow trout were carried out. Although the three probiotics showed inhibitory capacity and reduced the adhesion activity of *S. parasitica* cysts to cutaneous mucus in vitro, none of the three bacteria showed in vivo protection through either water or feed. The obtained results show the importance of selecting the most appropriate probiotic and its mechanism of action depending on the species of fish and the disease to be prevented.

**Abstract:**

Previous studies have demonstrated that the strains *Enterococcus gallinarum* L1, *Vagococcus fluvialis* L21 and *Lactobacillus plantarum* CLFP3 are probiotics against vibriosis or lactococosis in sea bass or rainbow trout. In this study, the utility of these bacterial strains in the control of saprolegniosis was evaluated. For this purpose, both in vitro inhibition studies and competition for binding sites against *Saprolegnia parasitica* and in vivo tests with experimentally infected rainbow trout were carried out. In the in vitro tests, the three isolates showed inhibitory activity upon mycelium growth and cyst germination and reduced the adhesion of cysts to cutaneous mucus; however, this effect depended on the number of bacteria used and the incubation time. In the in vivo test, the bacteria were administered orally at 10^8^ CFU g^−1^ in the feed or at 10^6^ CFU ml^−1^ in the tank water for 14 days. None of the three bacteria showed protection against *S. parasitica* infection either through water or feed, and the cumulative mortality reached 100% within 14 days post infection. The obtained results show that the use of an effective probiotic against a certain disease in a host may not be effective against another pathogen or in another host and that the results obtained in vitro may not always predict the effects when used in vivo.

## 1. Introduction

Saprolegniosis is a fungal disease, the main pathogenic agent of which is *Saprolegnia parasitica*, that can cause heavy economic losses in inland fish farms due to depletion of fish stock and eggs [1]. This disease affects freshwater fish at any stage of development and is characterized by skin lesions such as patches with a cotton wool-like appearance. Current prevention and treatment measures for saprolegniosis rely on the use of chemicals products that may cause damage to the environment [2,3] and humans [4]. In addition, none of the available compounds offers enough protection after the rearing period [5]. Recently, metal-based nanoparticles have attracted attention as a potential material for prevention and control of saprolegniosis [6,7]. Among the biological control alternatives to chemotherapy is the use of plant extracts [8,9,10,11,12] or probiotics [13,14].

In this sense, a number of in vitro studies have evidenced that some bacterial strains can inhibit *Saprolegnia* spp., including *Pseudomonas fluorescens* [15,16,17,18,19], *Alteromonas* sp., *Pseudomonas alcaligenes*, *Pseudomonas saccharophila*, *Aeromonas caviae* and *Aeromonas eucrenophila* [20], *Serratia marcescens* [21], *Bacillus subtilis* [22], *Aeromonas media* [23], *Aeromonas sobria*, *Pantoea agglomerans*, *Serratia fonticola*, *Xhantomonas reflexus* and *Yersenia kristensenii* [17,18], *Lactobacillus plantarum* [24] and *Pseudomonas aeruginosa* [25]. However, their action in vivo has only been studied in eel *Anguilla australis* [13], silver perch *Bidyanus bidyanus* [14] and eggs of rainbow trout *Oncorhynchus mykiss* [26] or eggs of gourami *Osphonemus gouramy* [27].

In previous studies by our research group, various bacterial isolates with an in vitro ability to inhibit the growth of *S. parasitica* were obtained [17]. Their harmless nature with respect to rainbow trout (*Oncorhynchus mykiss*) and their ability to adhere the cutaneous mucus and reduce the adhesion of zoospores and cysts of *S. parasitica* were also demonstrated [18]. It was also found that of the fifteen investigated isolates, under experimental conditions, two isolates of *Pseudomonas fluorescens* reduced infection by *S. parasitica* in rainbow trout when these bacteria were added to the water in the tanks [28], and the mode of action of these bacteria was likely associated with the production of siderophores [29].

*Enterococcus gallinarum* L1 (obtained from *Dicentrarchus labrax* gut content) and *Vagococcus fluvialis* L21 (obtained from *Solea solea* gut content) demonstrated high levels of protection against *Vibrio anguillarum* in European sea bass (*Dicentrarchus labrax*) after an experimental challenge when these bacteria were added to feed for 20 days [30,31]. In the same way, *Lactobacillus plantarum* CLFP-3 (obtained from *O. mykiss* cutaneous mucus) added to the diet of rainbow trout for 36 days was demonstrated to improve protection against *Lactococcus garvieae* [32]. In the present study, the in vitro inhibitory activity of these probiotic bacteria against *S. parasitica* was determined, and in vivo tests with experimentally infected rainbow trout were carried out in order to investigate their potential use for the biocontrol of saprolegniosis.

## 2. Materials and Methods

### 2.1. Collection of Isolates Used in the Study

Three probiotic bacteria (*Enterococcus gallinarum* L1, *Vagococcus fluvialis* L21 and *Lactobacillus plantarum* CLFP3) previously obtained by our research group that were stored at −80 °C in our laboratory facilities were used in this study (Table 1). The first two have shown their efficacy against infection by *Vibrio aguillarum* in sea bass (*Dicentrarchus labrax*), and *L. plantarum* reduced mortality in rainbow trout (*O. mykiss*) infected with *Lactococcus garvieae*.

After thawing, the bacterial isolates *E gallinarum* L1 and *V. fluvialis* L21 were cultured overnight at 20 °C in 3 mL of brain–heart infusion broth (BHIB, Pronadisa, Condalab, Madrid, Spain), while *L. plantarum* CLFP3 was cultured in Man, Rogosa and Sharpe broth (MRS, Pronadisa). An aliquot of 500 μL of the culture was inoculated into 25 mL of BHIB or MRS broth and incubated at 20 °C and 200 rpm on a rotary orbital shaker (Innova^®^ 44, New Brunswick Scientific, Edison, NJ, USA) until the middle of the exponential growth phase (based on previously established growth curves). Bacteria were pelleted and washed twice by centrifuging at 1000× *g* for 15 min and resuspended in sterile saline solution. The number of bacteria was adjusted to the desired concentration according to the experiment in which they were used (see below) by counting in a hemocytometer chamber (Improved Neubauer Type, Albert Sass, Germany).

The strain *Saprolegnia parasitica* TRU12 isolated from a wild brown trout (*Salmo trutta*) with saprolegniosis [33] and maintained refrigerated in our laboratory was used. It was cultured for 3 d at 20 °C on glucose peptone (GP) agar; then, autoclaved half hemp seeds of *Cannabis sativa* were placed on the edges of the colony, followed by an additional 24 h of incubation at 20 °C. To obtain zoospores, the half hemp seeds colonized by hyphae were placed in Petri dishes with filtered autoclaved river water. After 36–48 h at 20 °C, the water was filtered through Whatman 541 cellulose filter paper. The number of zoospores was then estimated using a Hawksley Cristalite BS 748 counting chamber.

### 2.2. In Vitro Inhibition Assays of the Probiotic Bacteria against S. parasitica

To determine the in vitro inhibition of the probiotic bacteria strains against *S. parasitica*, different assays were performed. All these experiments were conducted following the procedures described in a previous study [17].

#### 2.2.1. Inhibition of Hyphal Growth on Solid Media (Plate Assay)

This test was performed in with BHI agar (Cultimed, Panreac Química SLU, Barcelona, Spain), whereby each bacterium was streaked twice. This assay was performed in triplicate with incubations at 3, 5 or 7 d at 20 °C. Then, a 3 mm diameter block of agar with young hyphal tips of *S. parasitica* was placed between the 2 bacterial streaks and incubated for 3 d at 20° C, measuring the diameter of the *S. parasitica* colony. A negative control plate was used without bacterial streaks under the same conditions.

#### 2.2.2. Inhibition of Hyphal Growth from Colonized Hemp Seeds and Cyst Germination in Liquid Media

Bacterial dilutions, hemp seeds colonized by *Saprolegnia* and zoospore suspensions were prepared following the methods described above. The bacterial concentration was adjusted to 2 × 10^5^ cells mL^−1^, and, taking this as the first dilution, four 10-fold dilutions of bacterial suspension were prepared. The concentration of zoospores was adjusted to 4 × 10^4^ zoospores mL^−1^. For the inhibition of hyphal growth, duplicate wells of a 24-well tissue culture plate (Falcon, Corning, Glendale, AZ, USA) filled with 1 mL of each bacterial dilution, 1 mL of BHIB and a half hemp seed colonized by *S. parasitica* were dispensed into each well.

For the inhibition of zoospores/cyst germination, the same procedure was followed, with the only difference being that 0.5 mL of the zoospore suspension (4 × 10^4^ zoospores mL^−1^) was added in place of a colonized hemp seed, and 0.5 mL of each bacterial dilution was used instead of 1 mL.

In both assays, plates were incubated for 3 d at 20 °C, and bacteria and *S. parasitica* zoospore negative controls were included on each plate. The presence or absence of macroscopic or microscopic hyphal growth and germination of cysts were observed by a Nikon Diaphot inverted microscope.

#### 2.2.3. Assay to Evaluate Fungicidal Effects

The initial step was the same as that for the hemp seed test, but the plates were placed in an incubator at 20 °C for 1, 2 or 3 d. Following each incubation period, the hemp seeds from the wells where the growth of *S. parasitica* was inhibited were removed, washed 3 times with sterile distilled water and transferred to a Petri dish containing 20 mL of filtered and autoclaved river water with streptomycin (2 × 10^2^ mg L^−1^) and penicillin (2 × 10^5^ UI L^−1^). The plates were incubated for 20 d at 20 °C. To avoid false-positive results due to inhibitory effects of viable bacteria in the mycelium, only the results with no bacterial growth in water samples collected after 24 h were accepted. A negative control was achieved by incubating a hemp seed colonized by *S. parasitica* in water both with and without antibiotics. The bacteria were considered to have a fungicidal effect if no mycelial growth was observed after 20 d.

#### 2.2.4. Inhibition Assays with Bacterial Culture Supernatants

The supernatants used to perform this assay were prepared according to Lategan et al. [34] using BHIB for L1 and L21 or MRS for CLFP3 strains. This supernatant and three 2-fold serial dilutions were tested to analyze the ability of the supernatants to inhibit hyphal growth (hemp-seed test) and to evaluate the capacity of the supernatants to inhibit cyst germination (cyst test). In the hemp-seed test, 2 mL of each supernatant dilution and a half hemp seed colonized by *S. parasitica* were dispensed in duplicate into each well of a 24-well tissue culture plate. In the cyst test, 1 mL of each supernatant dilution and 1 mL of the zoospore suspension (4 × 10^4^ zoospores ml^−1^) were added to each well. Incubation times and observation procedures were the same as those described in Section 2.2.2. Negative controls with only culture medium were included in both assays.

### 2.3. Adhesion to Skin Mucus of the Probiotic Bacteria and Inhibition of S. parasitica Cyst Adhesion

The adhesion capacity of the probiotic strains L1, L21 and CLFP3 to cutaneous mucus of trout and its potential to hinder the adhesion of *S. parasitica* cysts under conditions of exclusion, competition and displacement were investigated using the methods described by Carbajal-González et al. [18].

Cutaneous mucus was obtained from four male brown trout (*Salmo trutta*) with an average body weight of 196.7 ± 39.5 g from a hatchery belong to the Regional Government of Castile and Leon by scraping the surface with a plastic spatula. The mucus was centrifuged twice at 12 000× *g* for 5 min at 4 °C to eliminate particulate and cellular debris. The protein concentration was adjusted to 0.05 mg ml^−1^ in PBS using Bradford reagent (Sigma-Aldrich, Merck Life Science, Madrid, Spain). The resulting mucus suspension was finally sterilized by UV light exposure for 30 min and stored in aliquots at −20 °C until use.

Bacteria and cysts of *S. parasitica* were stained with Syto 9^®^ green fluorescent nucleic acid stain (Invitrogen, Fisher Scientific, Madrid, Spain) at a concentration of 1 μL per 10^9^ bacteria in 1 mL and 1 μL per 10^5^ cysts in 1 mL.

#### 2.3.1. Adhesion to Skin Mucus of the Probiotic Bacteria

This assay was performed in 96-well black polystyrene plates (Costar, ImmunoChemistry Technologies, Davis, CA, USA) coated with cutaneous mucus. In the adhesion assay with the probiotic strains, wells without coating or coated with BSA (0.05 mg ml^−1^, Sigma-Aldrich) or mucin from swine stomach type II (MSS; 0.05 mg ml^−1^, Sigma-Aldrich) were used as controls for non-specific adhesion. To coat the plates, 25 μL of mucus, BSA or mucin and 75 μL of coating buffer (16.8 g sodium hydrogen carbonate, 21.2 g sodium carbonate per liter, pH 9.6) were added to each well and left overnight at 4 °C. Subsequently, the wells were washed with PBS-Tween 20 (0.1%), the fluorescently labelled bacterial solution (25 μL of 10^9^ bacterial cells ml^−1^) was added and plates were centrifuged at 163× *g* for 12 s to promote the adherence of bacteria to the mucus. Bacterial fluorescence was measured with a spectrophotometer (Synergy HT Multi-Detection Microplate Reader, Bio-Tek^®^, Winooski, VT, USA) at 485 nm excitation and 535 nm emission after 30 min of incubation in darkness and at room temperature (time 0). Non-attached bacteria were removed by washing with saline solution (50 µL/well), and plates were placed upside down on absorbent paper and centrifuged to remove any remaining liquid. Then, 50 µL of saline solution was added per well, and a new measurement was performed. Adhesion was expressed as the percentage of fluorescence recovered in the second measurement relative to the fluorescence at time 0.

#### 2.3.2. Inhibition of *S. parasitica* Cyst Adhesion

Seven serial 10-fold dilutions were performed from non-stained bacteria (2 × 10^9^ cells mL^−1^) in saline solution. The cysts were stained using the method mentioned previously using a concentration of 10^5^ cysts mL^−1^. In the exclusion test, each dilution of the bacterial isolates was added to the wells, and after being incubated and washed, the labelled cysts were added. In the competition test, both bacterial dilution and stained cysts were added simultaneously, and in the displacement test, the labelled cyst suspension was first added, incubated and washed; then, each dilution of the bacterial suspension was added. Plates were washed with saline solution after 60 min of incubation at room temperature for each condition; then, the fluorescence was recorded. Wells with saline solution instead of bacterial cells were used as controls in this assay. The percentage of cysts bound to these control wells was considered the reference value (100%), and the percentage of cyst adhesion in the presence of bacteria was compared to this reference value.

#### 2.3.3. Statistical Analysis

In the adhesion assays, statistical analyses were carried out using SPSS for Windows version 26 (IBM SPSS Statistic, Armonk, NY, USA) by the Student’s *t*-test (*p* ≤ 0.05). Data were shown as the mean ± standard deviation of three separate trials.

### 2.4. Pathogenicity for Rainbow Trout

The probiotic bacterium CLFP3 (*L. plantarum*), which has previously been used successfully to prevent lactococcosis in trout [32], was obtained from rainbow trout. However, the L1 (*E. gallinarum*) and L21 (*V. fluvialis*) bacteria were obtained from marine fish, and although they were previously reported to be non-pathogenic for sea bass [30,31], their potential pathogenicity for rainbow trout is unknown. Therefore, prior to being used in in vivo tests, experimental inoculations with L1 and L21 isolates were conducted to verify their lack of pathogenicity for rainbow trout.

Rainbow trout (*O. mykiss*) from a commercial fish farm with an average body weight of 37.24 ± 7.56 g were acclimatized for 10 days in a 120 L tank and observed daily to ensure that they did not show clinical signs of disease. Trials were conducted using groups of 20 fish kept in 40 L tanks filled with chlorine-free well water with a renewal rate of 1.5 L per h at 12 °C with constant aeration and a photoperiod of 12/12 h. Effluent water was disinfected with ozone. The experimental protocol was approved by the Subcommittee for Experimentation and Animal Welfare of the University of Leon, Spain (protocol number ULE_04_2015).

The bacterial isolates were grown in BHIB as described in Section 2.1 to obtain a suspension of 10^7^ cells mL^−1^. Fish were anaesthetized with tricaine methanesulfonate (MS-222, 50 mg mL^−1^) and injected with 0.1 mL intraperitoneally (ip) and intramuscularly (im) in two separate groups of 20 trout each. Control groups of 20 trout were inoculated with 0.1 mL of saline solution at the same time points. Fish were observed for 10 days and euthanized by overdosing them with MS-222 (100 mg mL^−1^) at the end of the observation period. They were subsequently examined for evidence of disease (i.e., gross lesions). Additionally, loopfuls of material from the kidney, liver and spleen were spread over plates of BHI agar with incubation at 20 °C for 3 days to determine the presence or absence of the inoculated bacterial isolates in the fish. The recovered bacterial isolates were examined by means of Gram-stained smears and basic biochemical tests such as catalase and oxidase tests.

Temperature and other physicochemical parameters of water were measured daily, and no significant differences were observed between the different groups (temperature, 12.47 ± 0.24 °C; pH 7.92 ± 0.06; dissolved oxygen, 9.72 ± 0.16 mg L^−1^; nitrites (NO_2_), 0.034 ± 0.005 mg L^−1^; and non-ionized ammonia (NH_3_), 0.059 ± 0.005 mg L^−1^).

### 2.5. Biocontrol of Saprolegniosis by Adding Bacteria to Water or Feed

#### 2.5.1. Biocontrol by Adding Bacteria to Water

For each of the tested bacteria, 60 rainbow trout with an average body weight of 28.79 ± 10.01 g in three 40 L tanks were used: 20 rainbow trout for treatment with the probiotic and 2 as controls for infection and ami momi treatment, respectively. Fish were infected with *S. parasitica* TRU12 following the method described previously by Fregeneda-Grandes et al. [33]. Briefly, the fish were slightly scarified on the skin using a variation of ami momi treatment [35]. Groups of 10 fish were shaken for 2 min in a cylindrical stainless-steel colander with a mesh of 7 mm before adding a suspension of *S. parasitica* zoospores to the water to obtain a final concentration of 3 × 10^2^ spores mL^−1^ in the tank water. At the same time, a bacterial suspension of 10^6^ cells ml^−1^ in PBS was added. During the next 48 h, the flow of water was halted and then resumed. The bacterial treatment was repeated every 24 h for 14 days, halting the flow of water for 6 h. The bacterial suspension was replaced by the same volume of sterile PBS in the infection control tank, while neither bacteria nor zoospores were added to the tank to control for death caused by ami momi treatment. The fish were monitored under observation for 14 days to detect the emergence of signs of saprolegniosis.

#### 2.5.2. Biocontrol by Adding Bacteria to Feed

To conduct these experiments, three groups of 20 rainbow trout with an average body weight of 29.29 ± 10.33 g were maintained in 40 L tanks: one tank for the bacterial treatment and two tanks as controls. Fish were fed for 14 days with T2 Optiline 1P (Skretting^®^, Burgos, Spain) feed containing 10^8^ bacteria g^−1^ feed, with a daily administration of 2% of tank biomass, followed by a fast of 48 h and 24 h before and after infection with *S. parasitica*, respectively. After infection, the feed with the bacteria continued to be administered for another 10 days. To prepare the feed with the probiotics, 10 mL of a bacterial suspension with a concentration of 10^9^ cells mL^−1^ was added to 90 g of feed, and the mixture was homogenized with an electric hand mixer (HM 3100, Braun GmbH, Germany) for 1 min. Subsequently, the feed was allowed to dry for 30 min at room temperature in a laminar flow cabinet. Infection with *S. parasitica* zoospores was performed as described above, and water flow was halted for the subsequent 24 h. The fish were maintained under observation to detect the emergence of signs of saprolegniosis.

The origin of the fish, their acclimation, experimental conditions and the quality of the physicochemical parameters of the water during these biocontrol tests were similar to those detailed in Section 2.4.

#### 2.5.3. Statistical Analysis

Survival analysis was performed by the Kaplan–Meier method, and the survival probability of the groups treated or not with the probiotics was compared using the log-rank test, considering that the differences were statistically significant at *p* ≤ 0.05. All calculations were performed using SPSS for Windows version 26.

## 3. Results

### 3.1. In Vitro Inhibition Assays of the Probiotic Bacteria against S. parasitica

#### 3.1.1. Assays with solid medium

The results of the plate assay in solid medium are shown in Table 2. On the negative control plates, the colony of *S. parasitica* reached more than 5 cm in diameter (Figure 1). All three bacteria showed a higher mycelial growth inhibition capacity when *S. parasitica* was tested on BHI medium with bacterial strains previously grown for 7 days. *L. plantarum* CLFP3 showed the highest fungistatic activity since it was already observed with a 3-day bacterial culture.

#### 3.1.2. Assays with Broth Medium (Hemp Seed Test and Cyst Test)

Both *E. gallinarum* L1 and *V. fluvialis* L21 isolates partially inhibited the growth of the mycelium of *S. parasitica* with the dilution containing 2 × 10^4^ bacteria mL^−1^ and with higher concentrations (Figure 1). On the contrary, *L. plantarum* CLFP3 did not inhibit the growth of the mycelium with any of the concentrations used.

*E. gallinarum* L1 and *V. fluvialis* L21 also inhibited cyst germination at concentrations greater than 4 × 10^3^ bacteria mL^−1^, but *L. plantarum* CLFP3 only controlled germination with the highest bacterial concentration of 4 × 10^5^ mL^−1^.

#### 3.1.3. Fungicidal Effect of the Bacteria

The three isolates showed to be lethal for *S. parasitica*, but the effect varied according to the concentration and incubation time of the bacteria. A lethal effect on the mycelium was achieved at lower bacterial concentrations as the incubation time increased. Thus, with one or two days of incubation, a lethal effect was observed only with the highest concentration (2 × 10^5^ bacteria mL^−1^), while with 3 days of incubation, a lethal effect was observed up to the third dilution (2 × 10^3^ bacteria mL^−1^). *V. fluvialis* L21 was the bacterium that presented the greatest fungicidal effect. An atrophied and small mycelium was observed after a period of incubation if a lethal effect was produced, while the mycelium grew and produced zoospores on the control plate and on the plate with a non-lethal concentration of bacteria.

#### 3.1.4. Assays with Supernatants

Culture supernatants only inhibited mycelium growth when they were undiluted, as well as in the case of *E. gallinarum* L1 with the first double dilution (1:2). Regarding the inhibition of cyst germination, none of the supernatants of the three bacteria was inhibitory.

### 3.2. Adhesion to Skin Mucus of the Probiotic Bacteria and Inhibition of S. parasitica Cyst Adhesion

Table 3 summarizes the results for the adhesion of the three probiotic strains to cutaneous mucus of brown trout. The adhesion was generally low (between 14.60% for *V. fluvialis* L21 and 7.54% for *L. plantarum* CLFP3), but it was higher than the adhesion of bacteria to bovine serum albumin and swine gastric mucus, although there were no significant differences between the adhesion of cysts to the mucus and that observed on the other substrates (*p* > 0.05, Student’s *t*-test).

The results of the exclusion, competition and displacement of *S. parasitica* cysts are shown in Table 4. In general, the three bacteria required lower concentrations to reduce the adhesion of cysts in the competition test (bacterial isolates and cysts were added at the same time). On the contrary, for the displacement of the cysts (bacterial isolates were added after the cyst), higher bacterial concentrations were required. *L. plantarum* CLFP3 was the bacterium that best reduced the adhesion of *S. parasitica* cysts in the three tests, despite presenting lower percentages of adhesion to mucus than L1 and L21 isolates (Table 3).

### 3.3. Pathogenicity for Rainbow Trout

Both *E. gallinarum* L1 and *V. fluvialis* L21 isolates were non-pathogenic for rainbow trout. No signs of disease or lesions were observed in any of the inoculated fish, and no mortality occurred during the days that the fish were kept under observation.

In the microbiological study of the internal organs of rainbow trout, only the injected strain was reisolated in two fish inoculated with *E. gallinarum* L1. In one of the fishes, which was inoculated intramuscularly, it was reisolated from the kidney, and in the other, which was injected intraperitoneally, it was detected in the spleen.

### 3.4. Biocontrol of Saprolegniosis by Adding the Bacteria to Water or Feed

None of the three probiotic strains, i.e., *E. gallinarum* L1, *V. fluvialis* L21 or *L. plantarum* CLFP3, was able to prevent *S. parasitica* infection in rainbow trout when added to tank water or administered through feed. Between 24 and 48 h after infection, the first macroscopic *S. parasitica* lesions were observed, while the first deaths were noted on day 3 post infection, reaching 100% mortality between days 6 and 9 post infection both in the groups treated with the probiotic strains and in those not treated (Figure 2), with no statistically significant differences observed. In the control groups (not infected with *S. parasitica*), 100% survival was observed, except in the experiment with the *E. gallinarum* L1 strain, in which one trout died 24 h post infection, probably due to the ami momi treatment.

## 4. Discussion

Previous studies have shown that *E. gallinarum* L1, *V. fluvialis* L21 and *L. plantarum* CLFP3 can be used as potential probiotics against vibriosis or lactococosis for sea bass or rainbow trout. In the present work, the possible utility of these bacteria in the control of saprolegniosis was studied. For this purpose, both in vitro inhibition studies and competition for binding sites against *S. parasitica* and in vivo tests with experimentally infected rainbow trout were carried out.

The three bacteria inhibited the growth of *S. parasitica* in solid medium, with the most pronounced inhibition in the case of *L. plantarum* CLFP3, since it appeared after three days of incubation of the bacteria. The greatest inhibition was observed in bacterial cultures that already had 7 days of growth. Our results agree with those obtained by Hussein and Hatai [20], although they found differences depending on the culture medium used; they observed the greatest inhibition when they used BHI agar with longer incubation times.

Regarding the inhibition of *S. parasitica* in liquid culture with hemp seed, the strains *E. gallinarum* L1 and *V. fluvialis* L21 prevented mycelial growth, while *L. plantarum* CLFP3 was unable to control it. However, the three bacteria inhibited the germination of cysts, although *L. plantarum* CLFP3 required higher concentrations. In all cases, the concentration of bacteria necessary to inhibit mycelial growth was greater than that required to prevent cyst germination. These data agree with those obtained by Bly et al. [16], who found that small amounts of different bacteria of the *Pseudomonas* genus prevented the germination of cysts. Carbajal-González et al. [18] suggested that the need for lower concentrations of bacteria to inhibit cysts may be due to the fact that a hyphal thallus already exists in hemp seeds, while the cysts have to germinate and form the mycelium. Inhibition of cyst germination in vitro could indicate that these bacteria hinder germination on the external surface of the fish, thereby helping to control *S. parasitica* infection. In addition, the three bacterial isolates were lethal to *S. parasitica*. This fungicidal effect is of interest since the three bacteria are able to inactivate *S. parasitica* and help control infection, acting not only as fungistatics.

None of the supernatants of the three bacteria inhibited the germination of *S. parasitica* cysts, in agreement with [16]; however, in our case, the supernatants did stop the growth of the preexisting mycelium in the hemp seed. The greater inhibition exhibited by *E. gallinarum* L1 may be due to the fact that this bacterium produces lactic and acetic acid, as well as small amounts of ethanol [31].

Epithelium surface colonization and adhesion, interfering with the pathogen’s adhesion, are two positives for the selection of candidate probiotics [36]. Adhesion to the cutaneous mucus of brown trout for the three bacterial strains tested in the present work was found to be low. The limited adhesion of bacteria to fish mucus agrees with results reported by other authors, with greater adhesion on the intestinal mucus of rainbow trout and other fish species than on cutaneous mucus [37,38]. Comparing the adhesion reported by Sorroza et al. [30,31] for *E. gallinarum* L1 and *V. fluvialis* L21 to the intestinal mucus of various marine fish and to the cutaneous mucus of brown trout obtained in the present study, both bacteria presented a higher percentage of adhesion to intestinal mucus. These results agree with those of Bálcazar et al. [38] who, using three bacteria from the intestinal microbiota of rainbow trout, observed a greater adhesion to intestinal mucus than to cutaneous mucus. The greater adhesion of *E. gallinarum* L1 and *V. fluvialis* L21 bacteria to intestinal mucus may be related to the origin of these bacteria, since they were isolated from intestinal mucus; however, *L. plantarum* CLFP3, which was originally isolated from cutaneous mucus, presented percentages of adherence to skin mucus lower than the other two bacteria. Chabrillón et al. [37] indicated no specific adhesion of bacteria to a specific host or substrate but that the adhesion capacity depends more on the strain. This is in agreement with the data obtained in the present study for *E. gallinarum* L1, *V. fluvialis* L21 and *L. plantarum* CLFP3, since there were no significant differences in adhesion to the different substrates used, although adhesion to skin mucus was greatest.

Although adhesion to skin mucus was low, the three bacteria isolates significantly reduced adhesion of *S. parasitica* cysts, although higher bacterial concentrations were required for displacement of *S. parasitica* cysts. This agrees with the results obtained by Carbajal-González et al. [18], who indicated that the greater need for bacteria for displacement may be due to a high adhesion of *S. parasitica* cysts to mucus, making it difficult for bacteria to subsequently eliminate them. The lowest concentrations of bacteria to significantly reduce attachment of *S. parasitica* cysts occurred under competitive conditions.

An essential characteristic of a potential probiotic is its safety and lack of pathogenicity. It has previously been shown that *L. plantarum* CLFP3 is safe for rainbow trout [32] and that *E. gallinarum* L1 and *V. fluvialis* L21 are safe for sea bass [30,31]. In the present study, found that L1 and L21 are also non-pathogenic for rainbow trout. The genera *Lactobacillus*, *Enterococcus* and *Vagococcus* are included in the group of lactic acid bacteria, which are mostly non-pathogenic and considered part of the commensal microbiota in the gut of several fish species. Thus, González et al. [39] reported the presence of *L. plantarum* and *V. fluvialis* as a part of the microbiota of several species of freshwater fish, including rainbow trout, and Petersen and Dalsgaard [40] isolated various *Enterococcus* spp., including *E. gallinarum*, from the intestine of fish from integrated chicken-fish farms. However, Osman et al. [41] found signs of septicemia in Nile tilapia (*Oreochromis niloticus*) associated with two isolates of *E. gallinarum*. In addition, other species of these genera, such as *V. salmoninarum*, have been isolated from diseased fish and are capable of causing mortalities up to 50% in rainbow trout farmed at low water temperature [42]. Regarding the genus *Enterococcus*, three species (*E. faecalis*, *E. faecium* and *E. hirae*) have been associated with enterococcosis/streptococcosis both in freshwater and marine fish [43,44]. Another *Enterococcus* species, *E. seriolicida*, initially described as pathogenic for yellowtail (*Seriola quinqueradiata*) in Japan [45] was later shown to be really identical to *Lactococcus garvieae* [46].

Although together, the three probiotic strains used in the present study achieved acceptable results in vitro, inhibiting the growth of *S. parasitica* and decreasing the ability of cysts to adhere to the cutaneous mucus, none of the three strains showed effective results in vivo to prevent experimental saprolegniosis in rainbow trout either through the water or administered in feed. Nevertheless, the beneficial effects against *S. parasitica*, even if limited to in vitro responses, must be considered since the combination of these bacteria with other strains could ensure a sufficient synergistic effect, with the capacity to prevent the presentation of the disease. Gram et al. [47] pointed out that the selection and application of probiotics must be tested for each individual host–pathogen combination and that in vivo activity cannot be predicted based on in vitro testing. These authors found that a strain of *Pseudomonas fluorescens* AH2 was strongly inhibitory against *Vibrio anguillarum* in vitro and that it was also able to reduce mortality in rainbow trout against *V anguillarum* infection via addition to the tank water [48]. Later, they found that this probiotic strain also inhibited *A. salmonicida* in vitro but was unable to prevent furunculosis in salmon (*Salmo salar*) in an in vivo model [47].

On the contrary, Nurhajati et al. [24] managed to prevent infection by *S. parasitica* in catfish (*Pangasius hypophthalamus*) by adding different concentrations of *L. plantarum* FNCC 226 to the water and found that the effect was dose-dependent, with higher concentrations of *S. parasitica* necessary to achieve inhibition. The best concentration treatment was between 4.2 × 10^5^ cfu mL^−1^ and 8.4 × 10^5^ cfu mL^−1^ of *L. plantarum* FNCC 226, which can inhibit infection with a concentration up 4 × 10^7^ zoospores mL^−1^ of *S. parasitica*. In our case, the concentration of *L. plantarum* CLFP3 used was very similar (10^6^ bacteria mL^−1^), but the concentration of zoospores was much lower (3 × 10^2^ zoospores mL^−1^), although the results are not necessarily comparable since a different experimental infection method was used, as well as different species and sizes of fish.

In previous works by our research group using the same experimental conditions, we observed that two *Pseudomonas fluorescens* strains (LE89 and LE141) were able to prevent infection by *S. parasitica* in rainbow trout when added to tank water but not when administered with feed [28]. Subsequently, we investigated the possible mechanisms of action of these two isolates and verified that they seems to be related to activity of competitive exclusion and to the production of some siderophores [29]. In addition, Lategan et al. [13,14] found that administration of *Aeromonas media* strain A199 in water significantly decreased the incidence of saprolegniosis in eel (*Anguilla australis*) and silver perch (*Bidyanus bidyanus*) and that the substance associated with the inhibitory activity was indole production [34]. In another study [49], the application of supernatant from a culture of a bacterium identified as *Burkholderia* sp. Reduced the rate of infection by *Saprolegnia* sp. in grass carp (*Ctenopharyngodon idella*). The substance involved with that antifungal activity was thermostable and was identified as 2-pyrrolidone-5-carboxylic acid, a derivate of glutamic acid.

Competitive exclusion is generally considered a major strategy by which probiotic bacteria prevent the adhesion of pathogenic microorganisms to fish mucous membranes. Although in vitro studies have frequently reported this effect, studies with animals have failed to reproduce the same results with in vivo models on many occasions [50], which could partly elucidate the results obtained in the present study.

Probiotics have been shown to have the capacity to increase innate and adaptive immunity of fish, with the effects exerted on the fish innate immune system as the main desirable characteristics [36]. Pérez-Sánchez et al. [51] investigated the expression of immune-related genes in rainbow trout feeding with *L. plantarum* CLFP3 (10^6^ CFU g^−1^ f for 36 days) and found that mRNA levels of IL-10, IL-8 and IgT were significantly higher in the *L. plantarum* group compared to the control group after *Lactococcus garvieae* infection, suggesting that protection conferred by *L. plantarum* was mediated by the stimulation of the immune response.

In the present study, the possible mechanism of action of the three isolates used was not investigated, so we do not know whether these isolates could increase the immune response of rainbow trout under the tested conditions. The mechanisms by which probiotics stimulate the immune system are not yet well understood, but it is known that factors such as the type of strain, dose, duration and mode of administration or environmental conditions can affect the immunomodulating potency of probiotics [52]. In this work, a dose of 10^6^ bacteria mL^−1^ in a water tank or 10^8^ bacteria g^−1^ for two weeks was used, which may not have been sufficient to stimulate the immune system. However, in the studies in which the usefulness of probiotic strains in the prevention of saprolegniosis has been confirmed, their beneficial effect seemed to be related to the production of inhibitory substances or competition for iron and not to the increase in the immune response [19,29,34,49].

## 5. Conclusions

The obtained results show that the use of an effective probiotic against a certain disease in a host may not be effective against another pathogen or in another host and that the results obtained in vitro may not always predict the effects when used in vivo. These results highlight the importance of selecting the most appropriate probiotic and its mechanism of action depending on the species of fish and the disease to be prevented.

## Figures and Tables

**Figure 1 animals-13-00954-f001:**
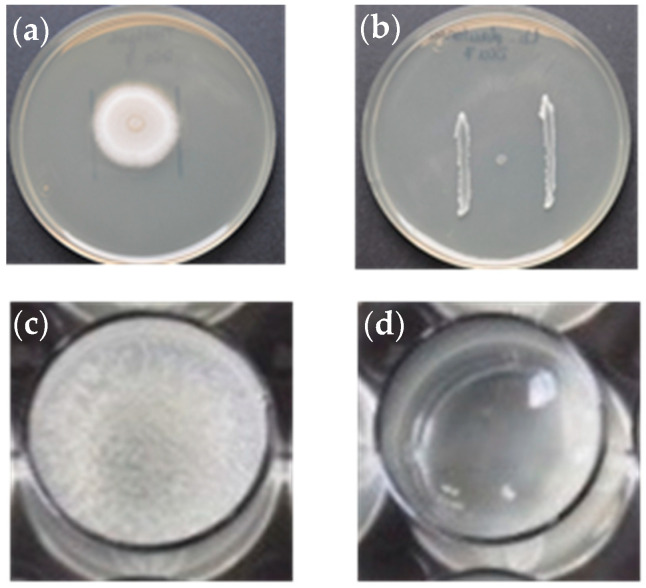
In vitro growth inhibition tests of the bacterial strains against *S. parasitica* on BHI agar (**a**,**b**) and in 24-well tissue culture plates with a colonized hemp seed and BHI broth (**c**,**d**): (**a**) *S. parasitica* after 3 days of incubation at 20 °C on the control plate; (**b**) high level of inhibition of *S. parasitica* by *Lactobacillus plantarum* CLFP3 strain after 3 days (bacterial strain was previously grown for 7 days at 20 °C); (**c**) *S. parasitica* after 3 days of incubation at 20 °C used as a control well; (**d**) partial growth inhibition of *S. parasitica* produced by *Enterococcus gallinarum* L1 strain (2 × 10^5^ cells mL^−1^) after 3 days at 20 °C.

**Figure 2 animals-13-00954-f002:**
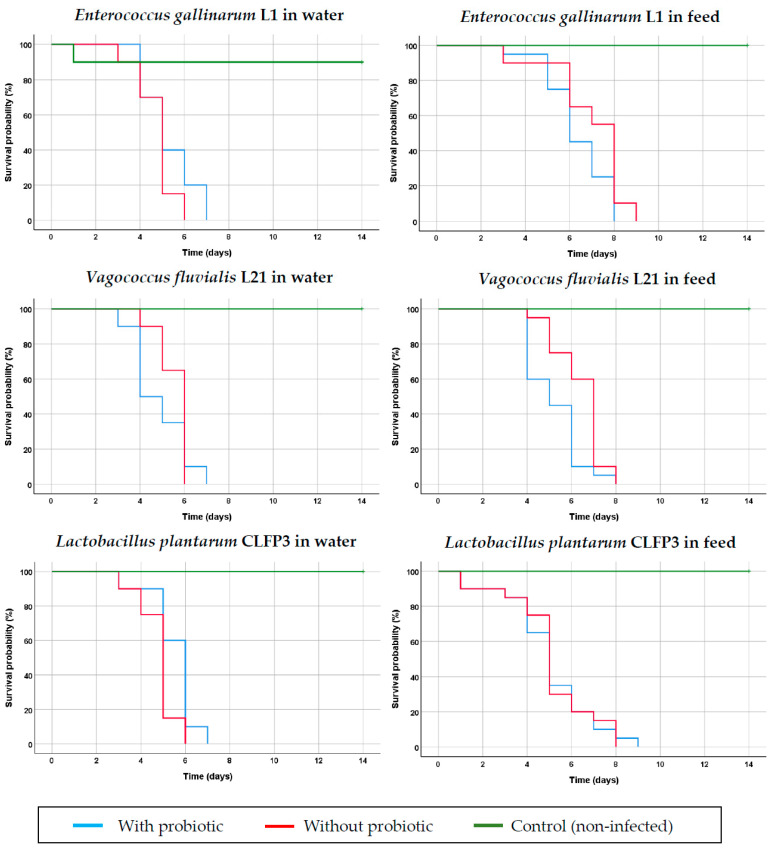
Kaplan–Meier survival curves in rainbow trout (*Oncorhynchus mykiss*) treated with the probiotics strains *Enterococcus gallinarum* L1, *Vagococcus fluvialis* L21 and *Lactobacillus plantarum* CLFP3 administered in water or in feed and infected with *Saprolegnia parasitica*.

**Table 1 animals-13-00954-t001:** Probiotic bacterial strains tested in the present study for biocontrol of saprolegniosis in rainbow trout (*Oncorhynchus mykiss*).

Bacterial Isolate	Origin	Probiotic Activity Against	Reference
*Enterococcus gallinarum* L1	Gut content of sea bass (*Dicentrarchus labrax*)	*Vibrio angillarum* in *D. labrax*	[31]
*Vagococcus fluvialis* L21	Gut content of sole (*Solea solea*)	*Vibrio angillarum* in *D. labrax*	[30]
*Lactobacillus plantarum* subsp. *plantarum* CLFP3	Cutaneous mucus of rainbow trout (*O. mykiss*)	*Lactococcus garvieae* in *O. mykiss*	[32]

**Table 2 animals-13-00954-t002:** Inhibition of *Saprolegnia parasitica* in solid medium (plate assay) shown as mean ± standard deviation (*n* = 3) colony diameter (cm) after culturing for 3 days at 20 °C on a BHI plate with a 3-, 5- or 7-day bacterial culture.

Incubation Time of the Bacteria (Days)	*Enterococcus gallinarum* L1	*Vagococcus fluvialis* L21	*Lactobacillus plantarum* CLFP3	Control Plate
3	4.25 ± 0.05	5.45 ± 0.04	0	5.70 ± 0.11
5	3.46 ± 0.06	2.90 ± 0.05	0	5.17 ± 0.07
7	0	2.35 ± 0.04	0	5.65 ± 0.14

**Table 3 animals-13-00954-t003:** Percentage of adhesion (mean ± SD of three experiments) of *Enterococcus gallinarum* L1, *Vagococcus fluvialis* L21 and *Lactobacillus plantarum* CLFP3 strains to cutaneous mucus of brown trout (CM), bovine serum albumin (BSA), mucin from swine stomach (MSS) and plate polystyrene (PP).

Bacterial Isolate	CM	BSA	MSS	PP
*Enterococcus gallinarum* L1	9.92 ± 7.59	4.57 ± 3.48	3.28 ± 3.48	4.39 ± 2.97
*Vagococcus fluvialis* L21	14.60 ± 5.48	7.87 ± 4.56	8.48 ± 3.89	7.98 ± 0.15
*Lactobacillus plantarum* CLFP3	7.54 ± 4.77	4.13 ±1.31	4.20 ± 1.45	4.51 ± 0.22

**Table 4 animals-13-00954-t004:** Percentage of reduction in the adhesion (mean ± SD of three experiments) of *Saprolegnia parasitica* cysts to cutaneous mucus of brown trout tested with different bacterial strain concentrations of *Enterococcus gallinarum* L1, *Vagococcus fluvialis* L21 and *Lactobacillus plantarum* CLFP3 under conditions of exclusion, displacement and competition.

Bacterial Strain	Bacterial Concentration	Exclusion *	Displacement *	Competition *
*Enterococcus gallinarum* L1	2.5 × 10^7^	**28.55 ± 0.01**	26.68 ± 2.92	65.03 ± 1.28
	2.5 × 10^6^	19.05 ± 6.73	**24.93 ± 5.97**	27.08 ± 3.05
	2.5 × 10^5^	7.50 ± 0.01	18.40 ± 14.77	6.33 ± 1.39
	2.5 × 10^4^	6.87 ± 1.89	0	**6.26 ± 1.22**
*Vagococcus fluvialis* L21	2.5 × 10^7^	83.11 ± 1.31	**93.24 ± 18.73**	73.86 ± 5.55
	2.5 × 10^6^	**62.81 ± 3.72**	25.09 ± 9.21	**40.03 ± 1.76**
	2.5 × 10^5^	55.45 ± 35.84	13.59 ± 9.88	7.39 ± 3.14
	2.5 × 10^4^	17.67 ± 22.71	6.45 ± 4.13	6.49 ± 4.15
*Lactobacillus plantarum* CLFP3	2.5 × 10^7^	87.50 ± 0.01	**33.32 ± 8.14**	33.33 ± 2.09
	2.5 × 10^6^	76.01 ± 6.48	22.30 ± 8.37	7.78 ± 1.45
	2.5 × 10^5^	**70.98 ± 8.39**	11.05 ± 8.33	5.44 ± 1.44
	2.5 × 10^4^	46.49 ± 17.63	0	**7.29 ± 0.68**

* Values in **blod** show the percentage of reduction in the adhesion of *S. parasitica* cysts significantly different (*p* < 0.05, Student´s *t*-test) from the control (no bacteria added) determined by testing the minimum effective numbers of bacteria.

## Data Availability

The data presented in this study are available upon request from the corresponding author.

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
