# Peer review of "Limited Probiotic Effect of Enterococcus gallinarum L1, Vagococcus fluvialis L21 and Lactobacillus plantarum CLFP3 to Protect Rainbow Trout against Saprolegniosis"

_animals, 2023, doi:10.3390/ani13050954_

Round 1

Reviewer 1 Report

The manuscript with ID (animals-2227966) by Fregeneda-Grandes and coauthors has evaluated the potential probiotic activities of Enterococcus gallinarum L1, Vagococcus fluvialis L21 and Lactobacillus plantarum CLFP3 against saprolegniosis in rainbow trout. The manuscript is interesting; however, several revisions should be addressed by the authors before the manuscript considered for publication in Animals.

Q1. Line 240: Why the authors used a dose of 107 cells /mL of the probiotic bacteria? Did the authors conduct LD50 testing prior to select this dose? This information missed in the manuscript.

Q2. Line 253: Please revise dissolved oxygen levels (9.72 ± 0.16 mg/L).

Q3. Line 264: Why the authors selected an infection dose of 3 × 102 spores /mL for experimental infection of rainbow trout?

Line 265: Similarly, why the authors used a water treatment with 106 of the probiotic bacteria /mL PBS ?

Q4. Line 275: How the authors included the probiotics in fish feeds with this dietary dose?

Q5. Line 276: 2% of the total biomass is insufficient for this rainbow trout size? Please revise.

Q6. Lines 285-286: Why the authors did not use Kaplan Meier analysis curves for evaluation of the fish survivals and cumulative mortalities?

Q7. Line 301: The resolution of Figure 1 should be edited? Please provide a high resolution one.

Q8. Results and Discussion, in general, missed important data that could support your results. I wonder why the authors did not evaluate the immune responses of fish in relation to dietary or water application of the probionts. These results could support your hypothesis and to reach to a possible attribution to stand over a solid ground about your results with possible suggestions for practical applications.

Author Response

Q1. Line 240: Why the authors used a dose of 107 cells /mL of the probiotic bacteria? Did the authors conduct LD50 testing prior to select this dose? This information missed in the manuscript.

Author´s answer: LD50 testing was not carried out; the dose of 107 cells/mL used was based on previous experiments by our group with these isolates (E. gallinarum and V. fluvialis) inoculated into sea bass and mice (Sorroza et al. 2012 and 2013; Doi:10.1016/j.vetmic.2011.09.013 and Doi:10.3147/jsfp.48.9) and other pathogenicity tests in rainbow trout with 21 bacterial isolates belonging to 6 different genera (Carbajal-González et al. 2013; doi:10.3354/dao02582).

Q2. Line 253: Please revise dissolved oxygen levels (9.72 ± 0.16 mg/L).

Author´s answer: This value is correct and is within the recommended range for rainbow trout. It must be considered that it is an average value of several measurements and that some of these measurements were made when the probiotic was administered and the water flow in the tanks remained closed, although with constant aeration (the levels were never less than 6 mg/l and ranged from 6.5 to 12.8 mg/l). These ranges are always recommended in the keeping conditions of this and other species.

Q3. Line 264: Why the authors selected an infection dose of 3 × 102 spores /mL for experimental infection of rainbow trout?

Author´s answer: This is the concentration of zoospores that our group has been using for a long time and that, together with the ami-momi method, guarantees a high percentage of infected fish. Initially, various concentrations of zoospores were tested with a high number of Saprolegnia isolates and this concentration was the best (Fregeneda-Grandes et al. 2001; doi:10.1046/j.1365-2761.2001.00305.x).

Q4. Line 275: How the authors included the probiotics in fish feeds with this dietary dose?

Author´s answer: 10 ml of a bacterial suspension with a concentration of 109 cells/mL in 90 g of feed were added and the mixture was homogenized with an electric mixer for 1 min. Subsequently, the feed was allowed to dry for 30 min at room temperature in a laminar flow cabinet. A brief explanation of the method has been included in the corresponding section (lines 269-273).

Q5. Line 276: 2% of the total biomass is insufficient for this rainbow trout size? Please revise.

Author´s answer: the amount of feed to be administered was suggested by the feed manufacturer (Skretting).

Q6. Lines 285-286: Why the authors did not use Kaplan Meier analysis curves for evaluation of the fish survivals and cumulative mortalities?

Author´s answer: as suggested by the reviewer, Kaplan Meier analysis  has been included (lines 280-283 in M&M and lines 359-365 in results section) and Figure 2 has been added to represent the survival curve (line 402).

Q7. Line 301: The resolution of Figure 1 should be edited? Please provide a high resolution one. 

Author´s answer: Low resolution images have been included in the manuscript to facilitate the submission and review process, but the original high resolution images will be provide.

Q8. Results and Discussion, in general, missed important data that could support your results. I wonder why the authors did not evaluate the immune responses of fish in relation to dietary or water application of the probionts. These results could support your hypothesis and to reach to a possible attribution to stand over a solid ground about your results with possible suggestions for practical applications.

Author´s answer: We agree with the reviewer´s observation, but evaluation of immune response of probiotiocs was not the aim of this work. As mentioned in the discussion, this paper has not investigated the possible mechanism of action in vivo of the probiotics used since none of the three seems to confer protection against infection by S. parasitica. However, in previous work by our group, we have obtained good results with two isolates of Pseudomonas fluorescens when they were added to the water (not in the feed) and it seems that the protective action would be due to the production of inhibitory substances by these bacteria and not to an immunomodulatory effect. These results coincide with those observed by other authors and appear as such in the discussion.

Reviewer 2 Report

The study and hypothesis are relevant to the scientific community and the aquaculture industry. The manuscript is well-written and easy to follow.

I just have some minor comments:

In the introduction, I think it’s important to say in the first paragraph that this is a fungal disease, or that Saprolegnia is a fungus.

From the MM, I can’t really understand where you got your isolates from. You say they were isolated from fish somewhere, but it is unclear if your group isolated them. Were they in stock at your lab facilities?  

L88: “reduced mortality in rainbow trout against infection” change to “reduced mortality in rainbow trout infected with”

L374: remove “been”

L375: change to “are potential probiotics” or “have probiotic activity”

L391-396: this sentence is too big, and the reader gets lost - divide it into 2 or 3 phrases

L480-482: in which fish species??

Author Response

Reviewer: In the introduction, I think it’s important to say in the first paragraph that this is a fungal disease, or that Saprolegnia is a fungus.

Authors: Added in line 44 that saproleniosis is a fungal disease as suggested by the reviewer.

Reviewer: From the MM, I can’t really understand where you got your isolates from. You say they were isolated from fish somewhere, but it is unclear if your group isolated them. Were they in stock at your lab facilities?

Authors: The three bacterial isolates as well as the S. parasitica strain were previously obtained by our group and are were kept frozen in our laboratory facilities. This section of the M&M section has been rewritten for clarity (lines 84-85).

Reviewer: L88: “reduced mortality in rainbow trout against infection” change to “reduced mortality in rainbow trout infected with”. Authors: Done (line 87).

Reviewer: L374: remove “been”. Authors: done.

Reviewer: L375: change to “are potential probiotics” or “have probiotic activity”. Authors: done.

Reviewer: L391-396: this sentence is too big, and the reader gets lost - divide it into 2 or 3 phrases. Authors: This part of the discussion has been rewritten for clarity.

Reviewer: L480-482: in which fish species?? Authors: fish species-rainbow trout-has been added (line 487).

Round 2

Reviewer 1 Report

The authors have properly addressed the comments raised by the anonymous reviewer